# Transcriptome Analysis of Differentially Expressed Genes Associated with Salt Stress in Cowpea (*Vigna unguiculata* L.) during the Early Vegetative Stage

**DOI:** 10.3390/ijms24054762

**Published:** 2023-03-01

**Authors:** Byeong Hee Kang, Woon Ji Kim, Sreeparna Chowdhury, Chang Yeok Moon, Sehee Kang, Seong-Hoon Kim, Sung-Hwan Jo, Tae-Hwan Jun, Kyung Do Kim, Bo-Keun Ha

**Affiliations:** 1Department of Applied Plant Science, Chonnam National University, Gwangju 61186, Republic of Korea; 2BK21 Interdisciplinary Program in IT-Bio Convergence System, Chonnam National University, Gwangju 61186, Republic of Korea; 3National Agrobiodiversity Center, National Institute of Agricultural Sciences, RDA, Jeonju 5487, Republic of Korea; 4SEEDERS Inc., Daejeon 34912, Republic of Korea; 5Department of Plant Bioscience, Pusan National University, Miryang 50463, Republic of Korea; 6Department of Bioscience and Bioinformatics, Myongji University, Yongin 17058, Republic of Korea

**Keywords:** cowpea, salt-stress, NGS, RNA sequencing, reference sequencing

## Abstract

Cowpea (*Vigna unguiculata* (L.), 2*n* = 22) is a tropical crop grown in arid and semiarid regions that is tolerant to abiotic stresses such as heat and drought. However, in these regions, salt in the soil is generally not eluted by rainwater, leading to salt stress for a variety of plant species. This study was conducted to identify genes related to salt stress using the comparative transcriptome analysis of cowpea germplasms with contrasting salt tolerance. Using the Illumina Novaseq 6000 platform, 1.1 billion high-quality short reads, with a total length of over 98.6 billion bp, were obtained from four cowpea germplasms. Of the differentially expressed genes identified for each salt tolerance type following RNA sequencing, 27 were shown to exhibit significant expression levels. These candidate genes were subsequently narrowed down using reference-sequencing analysis, and two salt stress-related genes (*Vigun_02G076100* and *Vigun_08G125100*) with single-nucleotide polymorphism (SNP) variation were selected. Of the five SNPs identified in *Vigun_02G076100*, one that caused significant amino acid variation was identified, while all nucleotide variations in *Vigun_08G125100* was classified as missing in the salt-resistant germplasms. The candidate genes and their variation, identified in this study provide, useful information for the development of molecular markers for cowpea breeding programs.

## 1. Introduction

Cowpea (*Vigna unguiculata* (L.) Walp.; 2*n* = 2x = 22) is a tropical herbaceous crop that has adapted to various abiotic stresses, including drought and heat stress [1,2]. Globally, the estimated area of cowpea cultivation is about 15 million hectares, with more than 8.8 million tons being produced annually. The whole of Africa occupies more than 95% of this cultivated area, especially the arid and semiarid regions of West Africa, which can be identified as the main cultivation areas for cowpea [3]. However, salt compounds in the soil of arid and semi-arid regions are generally not eluted due to the low frequency of rainfall [4], and the resulting accumulation of salt in the soil can increase the salt stress for cowpea and other important crops. This problem has been exacerbated by climate change, which has increased the rate of desertification and created larger areas of arid and semiarid land in West Africa [5,6], potentially reducing the yield and quality of crops, including cowpea in the region.

Salt stress causes various types of damage at all stages of a plant’s life cycle, from germination to seed production [7,8,9], with the proportion of cropland subject to salt damage reported to be increasing worldwide [10,11]. Therefore, understanding the effects of salt stress and the mechanisms associated with it is important from the perspective of meeting food demand in the future. Plants exposed to high soil salinity generally experience high osmotic and ionic stress, which affects a range of complex physiochemical processes [12,13,14,15]. The salinity reduces the leaf water potential and turgor pressure, leading to osmotic stress that induces abscisic acid biosynthesis, which in turn causes stomatal closure [16]. As a result, photosynthesis is reduced and oxidative stress increases [17]. In addition, excessive salinity around the roots can lead to ion toxicity, which increases the reactive oxygen species (ROS) levels, resulting in nutritional imbalances and damage to the cell structure [18]. This form of ion toxicity is commonly observed with sodium and chloride ions, which accumulate in highly saline soils [19].

One way to address high soil salinity is to create more salt-tolerant crop germplasms. However, salt stress is a complex process, and there are varying degrees of tolerance, both between and within species [20,21]. It has even been found that the response to salt stress can differ depending on the time of exposure and the stage of plant growth, with more rapid exposure resulting in more stress [22]. One study has reported that the difference in germination rates within a particular species ranged from 5.8% to 94.2% [23]. Within a particular species, individual plants with salt-sensitive genotypes tend to exhibit greater ion accumulation than salt-resistant genotypes do, leading to toxic effects [24]. These results suggest that salt tolerance is an independently evolved trait that can arise from completely unrelated mechanisms. This means that the genes associated with salt tolerance found in genetically distant species may not be effective if transplanted into germplasms of cowpea. However, genetic diversity within crops can be used to create germplasms with ideal traits, including salt tolerance [25]. Thus, further research is needed to identify genes related to salt stress in cowpea and to understand the mechanisms underlying their variation. In particular, it is important to understand plant ion homeostasis, osmotic responses, and oxidative stress in relation to increases in soil salinity.

Recently, the development of high-throughput sequencing technologies, such as next-generation sequencing (NGS), has made it possible to better understand plant genomes, which is essential for understanding complex traits, including those associated with salt stress [26]. For example, NGS-based RNA sequencing (RNA-seq) makes it possible to identify differentially expressed genes (DEGs) across the genome and analyze stress-related metabolic pathways via the functional annotation of the identified DEGs [27,28]. This approach provides an opportunity to search for candidate genes involved in the stress response of crops under salt stress, including the detection of rare transcripts, thus revealing the function and pathway of genes related to salt tolerance [29,30,31,32]. Though it is challenging to identify target genes from among the numerous DEGs generated via RNA-seq, reference genome information can be used to narrow down the range of candidate genes. As an example of this, kompetitive allele-specific PCR (KASP) genotyping assays have been widely used for SNP allele scoring and indel discrimination with various crops in a way that makes use of allele-specific primers [33]. For example, DEGs have been identified using RNA-seq in rices that differed in their salt tolerance, and SNP variation in the identified DEGs was then successfully used for KASP marker development [34]. These KASP markers can subsequently be employed for plant breeding through marker-assisted selection (MAS).

In the present study, we analyzed the expression patterns of genes associated with salt stress using cowpea germplasms with different levels of salt tolerance. We also conducted transcriptome profiling based on the fact that a plant absorbs salt from the roots, and that the damage is most severe at the seedling stage. This study thus aims to analyze the genetic network and related metabolic pathways for cowpea DEGs associated with salt tolerance.

## 2. Results

### 2.1. Physiological Responses to Salt Stress in Cowpea Germplasms

In this study, four cowpea germplasms with different levels of salt tolerance were exposed to a 250 mM NaCl solution for three weeks to simulate salt stress (Figure 1 and Figure 2).

Ion accumulation was generally higher in the salt-sensitive germplasms compared with the salt-resistant plants. In particular, the sodium and chloride ion levels for the salt-resistant germplasms (Vu_191 and Vu_328) were 27.65 mg/g and 82.12 mg/g, respectively, compared with 51.86 mg/g and 139.35 mg/g, respectively, for the salt-sensitive germplasms (Vu_393 and Vu_396).

### 2.2. Illumina Sequencing Pre-Processing, and Read Mapping

A total of 24 library samples were obtained for sequence processing, consisting of control (0 h) and NaCl treatments (24 h) for each of the four germplasms, with three replicates each. These library samples were sequenced using the Illumina Novaseq 6000 platform (Appendix A). RNA-seq analysis showed that the total number of clean reads generated for each sample was 1,144,868,572 (average length 101 bp). To obtain high-quality transcriptome short reads, bases with a Phred score (Q) of less than 20 were trimmed, and those trimmed reads with a length of less than 25 bp were eliminated. The total number of filtered reads was 1,107,552,070, with an overall average of 85.31% passing through the preprocessing stage, of which 1,038,301,592 (93.81%) were uniquely mapped to the cowpea reference genome sequence (Vunguiculata_540_v1.2). Of the 31,948 standard genes used for analysis, 27,559 had expression values and 25,476 had functional descriptions.

### 2.3. Identification of DEGs in Cowpea Germplasms with Different Salt Tolerance Levels

DEGs were screened using DESeq2 software based on a false discovery rate (FDR) of ≤0.01 and absolute values for the log_2_fold change (FC) of >1, with up-regulation defined as log_2_FC > 1 and down-regulation as log_2_FC < −1. The gene expression profiles of the cowpea germplasms with different salt tolerance levels were compared between the salt treatment and control samples (Figure 3, Appendix A). Overall, 5997 and 5532 DEGs were detected in the salt-resistant germplasms Vu_191 and Vu_328, respectively. The DEGs identified for Vu_191 typically included senescence-associated genes and genes encoding LEA proteins, while those identified for Vu_328 included genes encoding stress-induced proteins and pectin lyase. In addition, 5031 and 7444 DEGs were detected in the salt-sensitive germplasms Vu_393 and Vu_396, respectively. The DEGs identified for Vu_393 included genes encoding phosphatase family proteins and cytochrome P450 family proteins, while those identified for Vu_396 included nitrate transporters and auxin efflux carrier family proteins.

In addition, individual DEGs induced by salt treatment in four germplasms were compared (Appendix A). Overall, a higher number of up-regulated genes were identified in the salt-resistant germplasms, while the majority of the down-regulated genes were detected in the salt-sensitive plants. In the salt-resistant germplasms, 65 common DEGs, including LEA 4–5, were up-regulated, compared with 60 in the salt-sensitive germplasms, a group which included wall-associated kinase 3 (Figure 4a). In addition, 59 common DEGs, including metallothionein 2A, were down-regulated in the salt-resistant germplasms, compared with 99 for the salt-sensitive germplasms, including the cytochrome P450 family (CyP-89-A-5) (Figure 4b).

Gene ontology (GO) and Kyoto Encyclopedia of Genes and Genomes (KEGG) enrichment analyses were conducted in order to understand and classify the functions of the common DEGs identified for the different germplasms (Appendix A). In the salt-resistant germplasms, the up-regulated genes had nine enriched GO terms in the molecular function (MF) category, with many of the DEGs associated with catalytic and transferase activity. In addition, the down-regulated genes had only one enriched GO term in the MF category, with one DEG identified associated with ADP binding, unlike the up-regulated genes. On the other hand, there were no GO terms identified as being at a significant level (*p* < 0.05) from among the common DEGs for the salt-sensitive germplasms. KEGG analysis classified the common DEGs into five major groups for the salt tolerance germplasms. Most of the DEGs, for both salt tolerance levels, were associated with metabolism at the major classification and with the global/overview maps related to pathways or metabolism at the sub-classification.

### 2.4. Clustering Analysis of the Identified DEGs

Hierarchical clustering analysis was conducted to confirm the gene expression patterns using information from the 9784 DEGs that were significantly expressed for each salt tolerance germplasm (Appendix A). The identified DEGs were classified into six clusters containing 3710, 1762, 2961, 384, 276, and 691 genes, respectively (Figure 5).

For the C1 and C6 clusters, most of the DEGs were generally down-regulated while, for the C2 and C3 clusters, the DEGs were generally up-regulated across the four germplasms. Most of the relative expression levels were found to be similar in the four germplasms, but the C4 and C5 clusters exhibited distinct differences. The C4 cluster contained DEGs that were down-regulated in salt-resistant germplasm Vu_191 and those that were up-regulated in salt-sensitive germplasm Vu_396. This cluster contained DEG-encoding nodulin MtN21/EamA-like transporter family protein and NAD(P)-binding Rossmann-fold superfamily protein. On the other hand, the C5 cluster contained DEGs that were up-regulated in some salt-resistant germplasms and down-regulated in some salt-sensitive germplasms. This cluster contained DEG-encoding nitrate transporter 1.5 and the heavy-metal transport/detoxification superfamily protein.

GO and KEGG analysis was conducted to understand the functions of the DEGs included in each cluster (Appendix A). Overall, 384 DEGs in the C4 cluster were enriched for 29 GO terms, with 28 being independently classified into the MF category and 1 as a biological process (BP). In addition, many of the DEGs were associated with catalytic activity in the MF category, as is the case with the common DEGs, with a difference in that DEGs were detected for the response to oxidative stress in the BP category. However, in the C5 cluster, no GO terms were identified at a significant level (*p* < 0.05). As a result of the KEGG analysis of these genes, the C4 cluster was grouped into five major classifications and the C5 cluster into three (excluding genetic information and cell processing). In particular, metabolic terms were most common in both clusters, with sub-classifications dominated by global/outline maps related to pathways or metabolism, carbohydrate metabolism, and the biosynthesis of other secondary metabolites. However, this sub-classification had differences for six items, including membrane transport, transport, and catabolism.

qRT-PCR was employed to validate the expression of these DEGs (Appendix A). Of the DEGs with significant expression patterns, six were selected and analyzed further. The relative expression levels obtained from qRT-PCR exhibited trends similar to those of the Log2FC from RNA-seq.

### 2.5. Identification of Variations in the Target Gene

A total of 27 candidate genes were obtained based on the RNA-seq results, and two of these were not annotated (Table 1).

The 25 annotated genes included various genes related to salt stress, such as LATE EMBRYOGENESIS ABUNDANT PROTEIN 4–5, and POTASSIUM TRANSPORTER 6. Reference sequencing (re-seq) was conducted on the four germplasms to narrow down the candidate genes, and the results were integrated with those of RNA-seq. The whole genome re-seq results for each germplasm are presented in Appendix A. Of the 27 candidate genes, two containing variations that may be associated with salt resistance and sensitivity (*Vigun_02G076100* and *Vigun08G125100*) were selected (Figure 6).

*Vigun_02G076100* and *Vigun08G125100* were identified as encoding POTASSIUM TRANSPORTER 6 and EXOCYST COMPLEX PROTEIN EXO70, respectively. A total of eight coding SNPs (cSNPs) were found in the exons of these two genes. The five cSNPs found in *Vigun_02G076100* included three synonymous SNPs (sSNP) without amino acid substitutions and one synonymous variation without an amino acid substitution. However, the other SNP caused the substitution of lysine (Lys, K) in the positive amino acid with glutamic acid (Glu, E) in the negative amino acid when compared to the reference. This SNP was found in the salt-resistant germplasm Vu_191. The three cSNPs found in the other candidate gene *Vigun_08G125100* included two non-synonymous SNPs (nsSNPs). When compared with the reference, one SNP led to the substitution of aspartic acid (Asp, D) in the negative amino acid with histidine (His, H) in the positive amino acid, while the other SNP led to the substitution of glycine (Gly, G) in the special case amino acid with Asp. These two SNPs were found in both salt-sensitive germplasms Vu_393 and Vu_396. Interestingly, the salt-resistant germplasms Vu_191 and Vu_328, for which no SNPs were found, had a missing allele rather than the reference allele.

### 2.6. Validation of the Variations in the Candidate Genes

The variations in the two candidate genes were confirmed using DNA-seq and PCR analysis of the four cowpea germplasms. Of the cSNPs identified in *Vigun_02G076100*, one SNP, causing the substitution of another type of amino acid, was confirmed through DNA-seq. This was the same variation as observed in the four germplasms used for RNA-seq analysis, and the confirmed SNP was used to develop the KASP marker. In order to confirm that *Vigun_08G125100* was a missing allele, a primer producing a 1465 bp PCR product was designed (Appendix A). Interestingly, PCR products were only generated for this gene with the salt-sensitive Vu_393 and Vu_396. The variations in the two candidate genes were validated using a total of 20 cowpea germplasms that included the 4 germplasms used for RNA-seq (Table 2).

*Vigun_02G076100* was validated through the developed KASP marker. As a result, Vu_191 exhibited the same variation as seen in the re-seq analysis, and the SNP variation was observed in the salt-resistant germplasm Vu_111 (Appendix A). On the other hand, for *Vigun_08G125100*, PCR products were only generated for five salt-sensitive germplasms, including Vu_393 and Vu_396 (Appendix A), although this was not found in all salt-resistant cowpea germplasms.

## 3. Discussion

Cowpea is a legume crop that is widely grown in arid and semiarid regions because it is both heat- and drought-tolerant [1]. However, salt stress is becoming an increasingly serious issue for crops in these regions due to climate change [4]. In this study, we evaluated the ion accumulation response to salt stress using four cowpea germplasms with different levels of salt tolerance and conducted RNA-seq analysis (Figure 2). It was found that ion accumulation was significantly higher in the salt-sensitive germplasms rather than the salt-resistant germplasms. These results are in agreement with those reported by previous studies [19].

Of the four germplasms investigated in the present study, Vu_191 was classified as strongly salt-resistant, Vu_328 as weakly salt-resistant, Vu_393 as having a medium tolerance, and Vu_396 as a salt-sensitive germplasm. Based on these results, we screened candidate genes by focusing on DEGs with significant expression patterns in the comparison between Vu_191 and Vu_396.

The expression profiles of these DEGs were compared between control and treatment groups to identify genes associated with salt stress. Consequently, numerous DEGs related to salt stress were identified in the present study. For example, *Vigun_11G140800* encodes senescence-associated gene 12 (SAG 12), which is related to cysteine protease [35], and plays a role in plant aging and programmed cell death in response to biotic and abiotic stresses [36]. This was up-regulated in all four germplasms, suggesting that aging was accelerated by salt stress. In addition, *Vigun_09G159100* is a gene-encoding wall-associated kinase 3 (Wak3), which is associated with the pectin molecule in the cell wall and is essential for cell expansion [37]. It has been reported that a decrease in protein levels affects cell expansion and cell shape. In the present study, there was no significant expression value in the salt-resistant germplasms, but overexpression was observed in the salt-sensitive germplasms. This may be a form of plant defense to maintain homeostasis in response to stress such as excessive salt accumulation in salt-sensitive plants.

Another gene of note was *Vigun_03G195700*, which encodes CyP-89-A-5. The Cyp family is a large collection of proteins found in higher plants, and it has been assumed that they provide protection from various biotic and abiotic stresses. In particular, it has been reported that the suppression of CaCyP1 in pepper, which has a high homology with CyP-89-A-5 in *Arabidopsis*, increased the susceptibility to bacterial pathogens [38]. This gene is down-regulated in salt-sensitive germplasms, which is consistent with our results. Therefore, the Cyp gene found in cowpea is also assumed to affect salt tolerance via a similar mechanism.

We also conducted GO and KEGG analysis of the common DEGs and each cluster. The DEGs identified for the salt-resistant germplasms were generally related to catalytic activity (GO:0003824) and transferase activity (GO:0016740), while the DEGs corresponding to Cluster 4 were associated with small-molecule binding (GO:0036094), anion binding (GO:0043168), and ribonucleotide binding (GO:0032553). These GO terms play important roles in several salt tolerance mechanisms, including osmotic regulation. In particular, catalytic activity (GO:0003824) exhibited functions related to osmotic regulation and ionic change when exposed to salt stress [39]. These results also suggest that protein-coding genes, related to molecular structure and function, can be regulated in response to salt stress, and further indicate that anion reactions are associated with salt stress.

As a result of our KEGG analysis, most of the DEGs were associated with metabolism with four sub-classifications (global and overview maps, amino acid metabolism, carbohydrate metabolism, and biosynthesis of other secondary metabolites). This suggests that abiotic stress not only regulates metabolic processes via enzyme activity but also causes indirect or direct changes in proteins by affecting amino acids. Our results also suggest the involvement of the metabolism of various amino acids and the biosynthetic pathways of secondary metabolites such as phenylpropanoids. It has been reported that phenylpropanoids are activated under various abiotic stress conditions, including salt stress, to remove ROS [40]. These results thus help us to understand the molecular biological response to salt stress.

We subsequently selected 27 target genes, related to salt tolerance, based on the expression patterns and annotations for the DEGs in the RNA-seq analysis. One of these was *Vigun_01G124200*, which encodes LATE EMBRYOGENESIS ABUNDANT PROTEIN 4–5 (LEA 4–5). The LEA protein is a polypeptide that accumulates in later embryonic stages and is associated with the acquisition of desiccation tolerance [41]. It also increases resistance to osmotic and cold stress in various crops and is associated with water-deficient conditions [42]. This protein generally accumulates during periods of stress-induced growth arrest and is involved in stress recovery [43]. For example, AtLEA4–5 in *Arabidopsis* is known to be a member of the genes encoding the LEA protein involved in water deprivation tolerance [44]. This gene is usually suppressed by the repressor AtMYB44, but it has been reported that, when exposed to osmotic stress, the repressor is removed and normal expression occurs [45]. In the RNA-seq results, *Vigun_01G124200*, which encodes LEA4–5, was significantly up-regulated in the salt-resistant germplasms. However, the re-seq results did not detect significant variations. Interestingly, *Vigun_03G281700* encoding MYB44 was up-regulated in the salt-sensitive germplasms. This suggests that the expression of LEA4–5 in cowpea can be regulated by the same mechanism used in *Arabidopsis thaliana*, but that it is also regulated by an additional pathway.

Because interpreting the large volumes of data from the 27 target DEGs was difficult, we conducted re-sequencing to narrow down the range of the candidate genes. Most of the target genes had many SNPs in each salt-tolerant germplasm, but these SNPs were in the UTR or intron regions, which may not be involved in regulating gene expression. However, some of the SNPs in the two candidate genes, *Vigun_02G076100* and *Vigun_08G125100*, exhibited significant associations with salt tolerance. *Vigun_02G076100*, a gene-encoding POTASSIUM TRANSPORTER 6, was up-regulated in salt-resistant Vu_191. Potassium (K+) is an essential cation for plant growth and development and the regulation of enzyme activity, membrane potential, and turgor pressure [12]. High salinity is the result of the accumulation of excessive sodium (Na+) ions, which leads to ion stress. Plants are consequently unable to maintain K+ homeostasis, which ultimately adversely affects plant growth. Accordingly, one of the primary mechanisms associated with salt tolerance in plants is the maintenance of a balanced cation ratio in the cytoplasm [46]. In addition, the *Arabidopsis* KUP6 subfamily transporter is related to cell growth and potassium homeostasis and has been reported to be a major factor associated with osmotic control [47]. This suggests that the strong salt resistance of Vu_191 occurs as a result of the overexpression of potassium transporter 6. *Vigun_08G125100* encodes EXOCYST COMPLEX PROTEIN EXO70 and was up-regulated in both Vu_393 and Vu_396. The exocyst subunit EXO70 protein has been reported to be involved in anchoring and regulating membrane fusion and actin polarity in the plasma membrane of exocysts [48]. Some genes included in the exocyst gene family have been reported to be up-regulated with exposure to salt stress, but their exact functions have not been identified [49].

The variations in the two candidate genes were validated using KASP genotyping and PCR products. The cSNPs found in *Vigun_02G076100* were found in Vu_111 and Vu_191, both of which were salt resistant. This has been identified as a specific variation in some salt-resistant germplasms. Based on this, it can be assumed that Vu_111 had the same salt tolerance mechanism as Vu_191. Conversely, *Vigun_08G125100* was identified as a missing allele in NGS analysis. To validate these results, 20 cowpea germplasms were tested, with 75% classified as having the same salt tolerance type as before. Thus, it can be assumed that the loss of this gene has occurred as a result of the development of various salt resistance mechanisms, but the functional part has not been confirmed.

In summary, we identified two candidate genes related to salt tolerance that differed between cowpea germplasms with different levels of salt tolerance. These variations were developed as KASP and indel markers, respectively. The two developed markers thus have the potential to be useful molecular markers for the screening of germplasms in salt tolerance breeding programs.

## 4. Materials and Methods

### 4.1. Plant Material and Phenotyping of Salt Tolerance

In this study, 20 cowpea germplasms with different levels of salt tolerance (10 salt-resistant and 10 salt-sensitive) were used, and among them four showed distinct differences in salt tolerance under controlled conditions and were used for RNA-seq analysis (Appendix A). The 20 cowpea germplasms were then used to verify the SNP variation. The germplasm seeds were obtained from the Rural Development Administration (RDA) Genebank at the National Agrobiodiversity Center, Republic of Korea. The four cowpea germplasm were treated with 250 mM NaCl for seedlings in the V2 stage with the same growth after germination. After three weeks of NaCl treatment, the entire plant was sampled to evaluate the accumulation of sodium and chloride ions. The ion content was extracted from dried and pulverized leaf samples (150 mg) using 30 mL of distilled water for 1 h and then filtered through Whatman filter paper. The sodium ion levels were determined using a Na+ measuring instrument (Horiba, Kyoto, Japan), while the chloride ion levels were determined using an ion-selective electrode (Mettler Toledo, Columbus, OH, USA).

### 4.2. Salt Treatment

One hundred seeds from each germplasm were sterilized with 70% ethanol for 1 min and then washed with sterile water. The sterilized seeds were germinated in a plant growth chamber under long-day conditions (16 h light and 8 h dark), and similar seedlings were selected and transplanted into 1/2 Hoagland Nutrient Solution for hydroponic use. After two weeks of salt treatment, seedlings with the same growth were treated with 250 mM NaCl, while the control seedlings were placed in a solution without NaCl. After 24 h of salt treatment, the roots of the NaCl-treated and control seedlings were sampled. Each treatment and control group had three biological replicates, which were randomly sampled at 10 points and mixed into a single sample. The samples were frozen using liquid nitrogen and stored at −80 °C for use in subsequent experiments. Overall, a total of 24 RNA library samples were analyzed.

### 4.3. RNA Extraction, Construction of cDNA Libraries and Short Read Sequencing

Total RNA was extracted using an RNeasy Plant Mini Kit (Qiagen, Hilden, Germany). The quality and integrity of the extracted RNA were determined using a 2100 Bioanalyzer RNA instrument (Agilent, Santa Clara, CA, USA). Poly-A+ libraries were prepared using an Illumina Truseq Stranded mRNA Library Prep Kit (Illumina, San Diego, CA, USA), and the generated libraries were sequenced using an Illumina NovaSeq6000 platform. Both RNA extraction and cDNA library construction were conducted according to the manufacturer’s instructions.

### 4.4. Sequence Pre-Processing and Mapping of RNA-Seq Reads

In the sequenced transcriptome short reads, the adapter sequence was removed with cutadapt [50] and pre-processing was conducted using DynamicTrim and LengthSort in the SolexaQA package [51]. DynamicTrim removes low-quality bases at both ends of short reads to purify them, while LengthSort excludes trimmed reads of 25 bp or fewer from the analysis process. The clean trimmed reads were mapped onto the *Vigna unguiculata* (v1.2) reference genome from the Phytozome database (http: //phytozome.jgi.doe.gov/ (accessed on 1 December 2022)) using HISAT2 software [52]. HTSeq (v.0.11.0) [53] was used to measure expression as the total number of reads mapped to each gene. In order to avoid bias due to the germplasm in the sequencing numbers, normalization was conducted using the DEseq library [54].

### 4.5. Identification of Differentially Expressed Genes (DEGs)

DEGs were selected based on a twofold change in the number of mapped reads and an FDR of ≤0.01, with the adjusted *p* value calculated using Benjamini–Hochberg correction. Hierarchical clustering analysis was conducted using the amap [55] and gplot libraries [56] in R to determine gene expression patterns, which were calculated using Pearson’s correlation, and grouping was conducted through the complete method. GO enrichment was analyzed using reference GO information [57]. The significance level was set at 0.05 and the GO terms were classified into biological process (BP), cellular component (CC), and molecular function (MF) categories. Functional annotation was conducted for an e-value of ≤ 1 × 10^−100^ and best hits using amino acid sequences from the KEGG database [58] and BLASTP.

### 4.6. Quantitative Reverse Transcription PCR (qRT-PCR) for Validation of DEGs

First-strand cDNA was synthesized using SuperScript™ III First-Strand Synthesis SuperMix (Invitrogen, Waltham, MA, USA) following the manufacturer’s instructions. qRT-PCR was conducted on a StepOne Real-Time PCR System (Applied Biosystems, Foster City, CA, USA) using a Bio-Rad iQ™ SYBR Green Supermix Kit (Invitrogen, CA). The reaction mixture, containing 20 ng of cDNA, was analyzed according to the manufacturer’s instructions. The PCR conditions were as follows: holding, 1 cycle at 95 °C for 10 min; cycling, 40 cycles at 95 °C for 15 s and at 60 °C for 60 s. Then, the melting curve analysis was conducted to confirm the absence of a product and the dimer formation of the primers. The primers were designed using Primer3 software (v2.3.5) [59]. The CT values were normalized using the ubiquitin-conjugating enzyme E2 variant 1D (UE21D) gene stable under salt stress as a housekeeping gene [60] and gene expression was analyzed using the 2^−ΔΔCT^ method [61]. Three biological replicates were analyzed using the average of two technical replicates.

### 4.7. Whole-Genome Resequencing and DNA Sequencing

Genomic DNA was extracted using a DNeasy Plant Mini Kit (Qiagen) following the manufacturer’s instructions, and the integrity and purity of the extracted DNA samples were determined using 2.0% agarose gel and a Nanodrop ND 2000 spectrophotometer (Thermo Fisher Scientific, Waltham, MA, USA). The cDNA library was constructed and sequenced using the same NGS protocol as for RNA-seq. Paired-end reads were mapped onto the cowpea genomic reference genome and then entered into the nf-core/sarek’s analysis pipeline [62]. The DNA was sequenced using the PCR products of the candidate gene on an ABI 3730XL analyzer (Applied Biosystems). The primers used to generate the PCR products were prepared in the same way as the primers used for qRT-PCR. More detailed information on this process is provided in Appendix A.

### 4.8. Kompetitive Allele-Specific PCR(KASP) Primer Design and Validation

KASP primers were designed to detect SNP variation in the candidate genes according to the standard KASP protocol. Allele-specific primers included FAM (5′-GCTATAACCAGAACAGGCCATCTCAATTT-3′) and HEX (5′-TAACCAGAACAGGCCATCTCAA-TTC). The KASP primers were used to genotype the 20 cowpea germplasms using StepOnePlus software (Applied Biosystems). Genotyping was conducted using a mixture consisting of 50 ng/5 uL of DNA, 0.14 uL of KASP assay mix, and 5 uL of KASP master mix. The KASP cycling conditions were as follows: pre-PCR reading, 1 cycle at 30 °C for 1 min; holding, 1 cycle at 94 °C for 15 min; cycling, 10 cycles at 94 °C for 20 s and 61–55 °C for 1 min (reduction of 0.6 °C per cycle), and 26 cycles at 94 °C for 20 s and 55 °C for 1 min; and post-PCR reading at 30 °C for 30 s.

### 4.9. Statistical Analysis

Statistical analysis was conducted using analysis of variance (ANOVA) and least significant difference (LSD) tests in SPSS 27 (IBM, Armonk, NY, USA), with *p* < 0.05 employed to determine statistically significant differences between groups.

## 5. Conclusions

Four cowpea germplasms with different levels of salt tolerance were used to investigate transcriptome variations in roots under salt stress. RNA-seq analysis of the salt treatment and control groups, with three biological replicates assessed for each germplasm, led to the selection of 27 candidate genes related to salt stress. Of these, two candidate genes with significant variation were investigated further in this study. The two candidate genes contained cSNPs in the exon region and represented a missing allele, respectively. The information provided on the two candidate cowpea genes in relations to salt stress and presented in the present study has the potential to be used for genetic improvements in cowpea breeding programs.

## Figures and Tables

**Figure 1 ijms-24-04762-f001:**
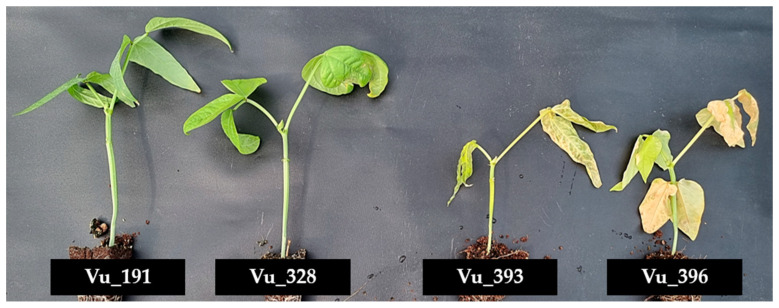
Four cowpea germplasms with different levels of salt tolerance exposed to a 250 mM NaCl solution for three weeks.

**Figure 2 ijms-24-04762-f002:**
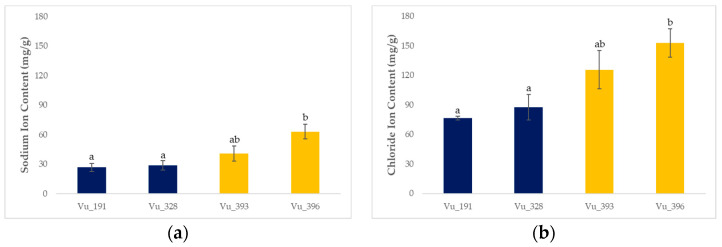
Ion accumulation for four cowpea germplasms exposed to 250 mM NaCl for three weeks: (**a**) sodium ions and (**b**) chloride ions. Different lowercase letters indicate statistically significant differences (LSD tests; *p* < 0.05). The error bars represent the standard deviation for five biological replicates.

**Figure 3 ijms-24-04762-f003:**
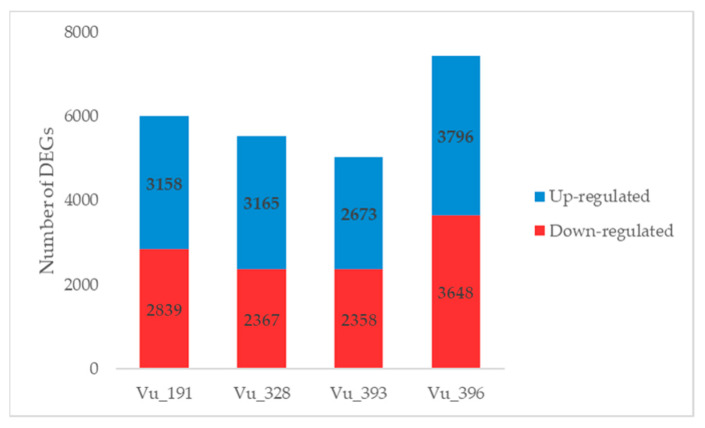
Number of DEGs identified in a comparison between the control and NaCl treatment for cowpea germplasms with different levels of salt tolerance.

**Figure 4 ijms-24-04762-f004:**
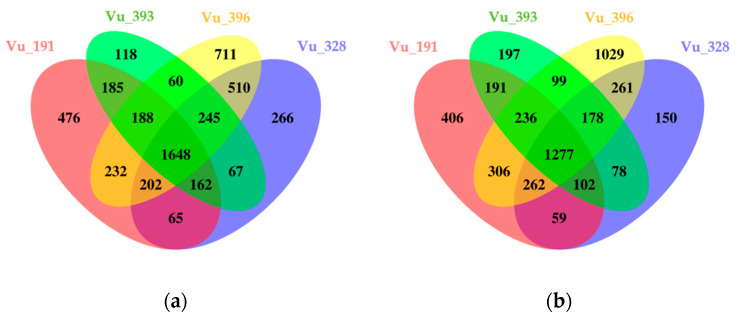
Venn diagrams showing the number of common and specific DEGs identified in cowpea germplasms with different levels of salt tolerance: (**a**) up-regulated and (**b**) down-regulated DEGs.

**Figure 5 ijms-24-04762-f005:**
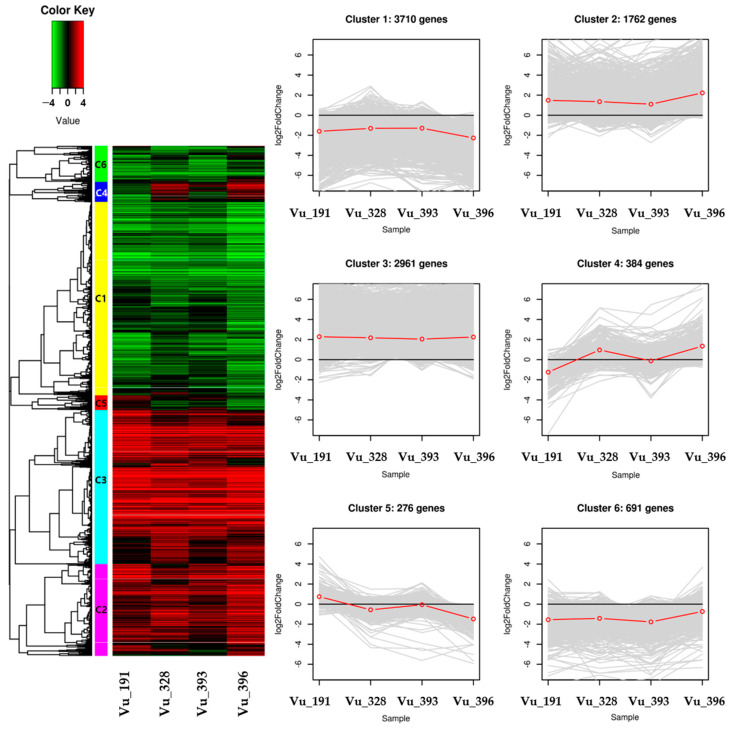
Heat map of the differential expression level of the genes in each cluster and line plots displaying the expressed clusters as patterns. The redline in line plots indicates the mean log_2_foldchange (FC) value in gene expression.

**Figure 6 ijms-24-04762-f006:**
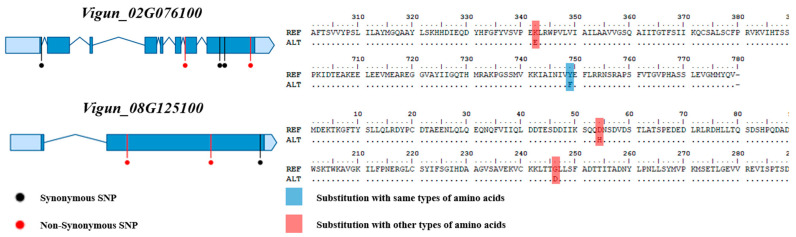
Gene model and variation information of two candidate genes related to salt-stress.

**Table 1 ijms-24-04762-t001:** Log_2_foldchange (FC) and functional annotations for differentially expressed genes (DEGs) with significant expression patterns. The color of background indicates the gene expression level from red (up-regulated) to green (down-regulated).

Gene id	Position	Log_2_FoldChange (FC) for Identified DEGs				Annotation
		Vu_191	Vu_328	Vu_393	Vu_396	
Vigun01g124200.1	30,107,016–30,107,917	3.71	2.51	−0.04	0.66	LATE EMBRYOGENESIS ABUNDANT PROTEIN 4–5
Vigun02g076100.1	22,812,209–22,818,056	3.35	−0.53	0.03	−0.91	POTASSIUM TRANSPORTER 6
Vigun02g087000.1	24,173,651–24,175,542	−2.55	−1.58	−1.40	1.73	ALCOHOL DEHYDROGENASE-RELATED
Vigun02g150700.1	29,780,314–29,782,539	3.70	0.77	0.71	−1.12	N-TERMINAL ACETYLTRANSFERASE
Vigun02g156800.1	30,300,350–30,306,429	4.74	−0.79	0.76	−2.85	OLIGOPEPTIDE TRANSPORTER-RELATED
Vigun03g036000.1	2,770,941–2,774,105	0.19	−0.98	−1.24	−2.64	RING-H2 FINGER PROTEIN ATL69-RELATED
Vigun03g195700.1	27,574,151–27,575,892	−0.49	−0.86	−1.45	−2.13	CYTOCHROME P450 89A2-RELATED
Vigun03g323700.1	51,942,392–51,948,451	3.10	−0.56	0.25	−2.84	ANION EXCHANGE PROTEIN
Vigun03g411400.1	61,890,298–61,892,363	−1.54	0.18	−0.65	1.31	PEROXIDASE 40
Vigun06g049900.1	17,502,581–17,504,135	1.89	1.00	0.25	−2.16	Hydroxycinnamate 4-beta-glucosyltransferase
Vigun06g206600.1	32,059,029–32,059,206	0.84	−0.07	−1.34	−1.61	Unknown
Vigun07g005500.1	439,864–440,793	3.34	1.49	1.55	−2.61	Hemopexin
Vigun07g044100.1	4,471,515–4,474,926	−2.91	0.52	0.32	2.81	3-oxoacyl-[acyl-carrier-protein] reductase
Vigun07g065400.1	7,663,218–7,665,263	−0.93	−0.34	2.22	1.80	Myb/SANT-like DNA-binding domain (Myb_DNA-bind_4)
Vigun07g164700.1	27,692,362–27,695,281	0.64	−3.29	−4.63	−4.02	ALUMINUM-ACTIVATED MALATE TRANSPORTER 10
Vigun07g217900.1	33,993,108–33,994,160	0.14	−4.35	−4.41	−5.92	Uncharacterized membrane protein
Vigun08g025300.1	2,190,472–2,192,518	−0.24	−1.85	−0.80	−3.35	HEAT STRESS TRANSCRIPTION FACTOR B-4
Vigun08g090000.1	20,808,945–20,813,724	4.14	−0.69	−0.46	−1.53	EamA-like transporter family (EamA)
Vigun08g116200.1	28,318,383–28,321,227	2.68	−1.19	−0.60	−2.22	COPPER TRANSPORT PROTEIN ATOX1-RELATED
Vigun08g125100.1	29,512,430–29,514,230	0.25	0.42	5.48	7.06	EXOCYST COMPLEX PROTEIN EXO70
Vigun09g086300.1	11,333,702–11,336,177	−2.42	1.31	2.73	3.33	INACTIVE POLY [ADP-RIBOSE] POLYMERASE SRO4-RELATED
Vigun10g015100.1	1,666,906–1,668,606	1.63	−0.20	−1.32	−1.96	CYCLIN-U4-1
Vigun10g180000.1	39,812,625–39,813,195	−0.79	0.97	1.66	2.02	Cotton fiber expressed protein (DUF761)
Vigun11g017700.1	2,206,680–2,207,648	−1.29	0.42	2.80	1.52	Unknown
Vigun11g018800.1	2,332,367–2,339,763	−0.10	−0.35	1.67	1.88	protein regulator of cytokinesis 1 (PRC1)
Vigun11g126800.1	33,386,966–33,389,552	1.96	−0.74	0.59	−2.28	MYB FAMILY TRANSCRIPTION FACTOR-RELATED
Vigun11g182400.1	38,574,316–38,575,114	0.20	−0.76	−1.17	−1.24	SAUR family protein (SAUR)

**Table 2 ijms-24-04762-t002:** Validation and comparison of the variation in the two candidate genes using 20 cowpea germplasms.

Sample	Salt Tolerance Type	*Vigun_02G076100*	*Vigun_08G125100*
Vu_035	Sensitive	REF (AAA, K)	Existence
Vu_266	Sensitive	REF (AAA, K)	-
Vu_296	Sensitive	REF (AAA, K)	Existence
Vu_318	Sensitive	REF (AAA, K)	Existence
Vu_319	Sensitive	REF (AAA, K)	-
Vu_343	Sensitive	REF (AAA, K)	-
Vu_348	Sensitive	REF (AAA, K)	-
Vu_393 *	Sensitive	REF (AAA, K)	Existence
Vu_396 *	Sensitive	REF (AAA, K)	Existence
Vu_403	Sensitive	REF (AAA, K)	-
Vu_055	Resistant	REF (AAA, K)	-
Vu_095	Resistant	REF (AAA, K)	-
Vu_111	Resistant	SNP (GAA, E)	-
Vu_129	Resistant	REF (AAA, K)	-
Vu_147	Resistant	REF (AAA, K)	-
Vu_166	Resistant	REF (AAA, K)	-
Vu_191 *	Resistant	SNP (GAA, E)	-
Vu_328 *	Resistant	REF (AAA, K)	-
Vu_336	Resistant	REF (AAA, K)	-
Vu_352	Resistant	REF (AAA, K)	-

* Four cowpea germplasms (Vu_191, Vu_328, Vu_393, and Vu_396) used for RNA-seq analysis.

## Data Availability

Not applicable.

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
