# Peer review of "Transcriptome Analysis of Differentially Expressed Genes Associated with Salt Stress in Cowpea (Vigna unguiculata L.) during the Early Vegetative Stage"

_ijms, 2023, doi:10.3390/ijms24054762_

Round 1

Reviewer 1 Report

To screen the salt stress-related candidate genes from cowpea is very important. Based on the comparative analysis of cowpea transcripts with different salt tolerance, two candidate genes which might be related to salt tolerance were inferred. But there are following deficiencies:

1) The research must verify the function of the two candidate genes. By now, there are not enough evidenc directly to verify the two genes of cowpea related to the salt tolerance.

2) In the result of 2.1(From the Line 101 to 115 ), the salt treatment time must be clear stated. 

3) Vu_111 is not the research material, but the statement from the Line 349 to 351 appeared the Vu_111.

4) The discription from the Line 364 to 367 is not related to the manuscript.

5) The format of the references needs correct.

Author Response

Hello,

First of all, We are really glad that you reviewed our paper.

We have corrected some sentences based on your advice.

Many Thanks

Reviewer 2 Report

The authors investigated 4 cowpea lines with different salt tolerance and detected 27 differently expressed genes related to salt stress and selected among them 2 candidate genes. The manuscript is well -written. The research is designed well, the methods are described in details. All together is a full and complete story with possible future plan.

Author Response

Hello,

We are really glad that you reviewed our paper.

Thank you for your evaluation and consideration.

Reviewer 3 Report

The manuscript “Transcriptome analysis of ifferentially expressed genes associated with salt stress in cowpea (Vigna unguiculata L.) during the early vegetative stage” is very intresting but required following revision.

Line-19. The authors used the “resistant” term against the abiotic stresses. It would be better to write as “tolerant”.

Line 29. It would be moreclear if authors add the nucleotide variation.

Line-39. Correct the sentence.

Line-47. Delete the “can”. because authors already cited the reference.

Line-52. Delete the “for example”.

Section 4.6. The qRT PCR progrmme is missing, need to add for readers.

In Fig-S1. It required revision. qRT PCR analysis was done using both control and stressed condition for its validation. The significance test is also missing.

In fig-5. The heat map is prepared only with the stressed data. The expression value of control data is not prsented.

Author Response

Hello,

First of all, We are really glad that you reviewed our pape.

We have corrected some sentences based on your advice.

Many Thanks

Round 2

Reviewer 3 Report

Now, The manuscript is ok.